# Corrosion Resistance Measurement of 316L Stainless Steel Manufactured by Selective Laser Melting

**DOI:** 10.3390/ma14164509

**Published:** 2021-08-11

**Authors:** Rigoberto Guzmán-Nogales, Francisco Estupiñán-López, Citlalli Gaona-Tiburcio, Omar E. Lopez-Botello, Juan G. Ramírez-Rodríguez, Patricia C. Zambrano-Robledo

**Affiliations:** 1Centro de Innovación e Investigación en Ingeniería Aeronáutica, Facultad de Ingeniería Mecánica y Eléctrica, Universidad Autónoma de Nuevo León, San Nicolás de los Garza 66455, NL, Mexico; rigo.guzmann@gmail.com (R.G.-N.); festupinan.ciiia@gmail.com (F.E.-L.); citlalli.gaona@gmail.com (C.G.-T.); juan_1979g@hotmail.com (J.G.R.-R.); 2Laboratorio Nacional de Manufactura Aditiva y Digital, Apodaca 66629, NL, Mexico; omarlopez@tec.mx; 3Tecnologico de Monterrey, Escuela de Ingenieria y Ciencias, Monterrey 64700, NL, Mexico

**Keywords:** 316L stainless steel, selective laser melting, electrochemical properties, linear polarization resistance, electrochemical noise

## Abstract

Selective laser melting (SLM) technology is ushering in a new era of advanced industrial production of metal components. It is of great importance to understand the relationship between the surface features and electrochemical properties of manufactured parts. This work studied the influence of surface orientation on the corrosion resistance of 316L stainless-steel (SS) components manufactured with SLM. The corrosion resistance of the samples was measured using linear polarization resistance (LPR) and electromechanical noise (EN) techniques under three different environments, H_2_O, 3.5 wt.% NaCl, and 20% H_2_SO_4_, analyzing the horizontal (XY) and vertical (XZ) planes. The microstructure and morphology of the samples were obtained by optical (OM) and scanning electron microscopy (SEM). The obtained microstructure showed the grains growing up from the fusion line to the melt pool center and, via SEM-EDS, the presence of irregular and spherical pores was observed. The highest corrosion rate was identified in the H_2_SO_4_ solution in the XZ plane with 2.4 × 10^−2^ mm/year and the XY plane with 1.31 × 10^−3^ mm/year. The EN technique along with the skewness factor were used to determine the type of corrosion that the material developed. Localized corrosion was observed in the NaCl electrolyte, for the XY and XZ planes (−1.65 and −0.012 skewness factors, respectively), attacking mainly the subgrains of the microstructure and, in some cases, the pores, caused by Cl ions. H_2_O and H_2_SO_4_ solutions presented a uniform corrosion mechanism for the two observed orientations. The morphology identified by SEM was correlated with the results obtained from the electrochemical techniques.

## 1. Introduction

Additive manufacturing (AM) technologies, considered as disruptive technologies [1], are capable of processing an extensive variety of advanced metallic alloys (e.g., aluminum, steel, nickel, and titanium based) [2,3]. AM technologies have gained the attention from various industrial fields (e.g., energy, automotive, and aerospace) thanks to their ability to produce high-complexity parts directly from 3D virtual models [4,5,6]. Some of the AM technologies commonly used to manufacture metallic parts are: selective laser sintering (SLS) [7], direct laser deposition (DLD) [8], electron beam melting (EBM) [9], and selective laser melting (SLM) [10]. The present research focused on the selective laser melting (SLM) technology, which is a powder bed fusion process that uses a high-power laser to melt and fuse metallic powders deposited layer by layer on a substrate [5,11,12,13]. In this process, optimal processing parameters such as laser power, scan speed, hatch spacing, and layer thickness are required in order to build fully dense parts.

316L stainless steel (SS), a typical alloy used in food processing, marine, chemical, and petrochemical processing, and medical device applications, is one of the most used metal alloys in SLM [14]. Properties such as strength, work hardening, and corrosion resistance are enhanced when manufacturing a part using SLM [11,14,15]. Nevertheless, the electrochemical response has not yet been fully determined; limited research about the corrosion behavior related to the morphology of SLMed 316L steels can be found [16,17,18,19]. In fact, Kong et al. [17] emphasized that the SLM defects such as melt pool boundaries and nonequilibrium phases are some of the main factors that decrease the corrosion resistance of an additively manufactured 316L SS compared with those of traditionally manufactured 316L SS. Nie et al. [20] established that the chemical composition of the passive film formed in an AM 316L SS plays an important role in its corrosion resistance. A minor content of hydroxide in the film combined with a micro-galvanic corrosion effect at the interface of the subgrain boundaries and subgrains reduce the potential to repair such a passive film, leading to the development of pitting corrosion. Zhou et al. [21] found that the melt pool boundaries of an SLM 316L SS considerably lead to pitting-corrosion attacks, but it can be significantly prevented by applying recrystallization post-processes, which help to remove the fusion lines produced by the laser process.

Another important study was recently made by Duan et al. [22] who found that the SLM 316L SS alloy tested in aggressive solutions (1 M of NaCl, pH 1, and 3 M of NaCl, pH 3) exhibits pitting corrosion at the gas pores induced by the melting process. In a similar background, the physical properties of the SLM parts also mainly depend on the microstructure orientation and the surface texture. For instance, the mechanical properties of an SLM material are better in the horizontal direction than in the vertical one [23]. According to Serafin et al. [24], the surface roughness of an SLM material also plays an important role, i.e., the corrosion resistance is better in an SLM surface after polishing than unpolished. This is related to the minimum number of cavities, defects, etc. on a polished surface, preventing the development of localized corrosion. Based on this, Geenen et al. [25] also found that the SLMed 316L SS does not show good electrochemical properties compared to specimens processed by casting, hot isostatic pressing (HIP), and SLM + HIP, due to the inhomogeneities caused by the nature of the manufacturing process, reducing the corrosion resistance of the material. In addition to these important findings, the amount of chromium on the surface of a 316L SS alloy influences its corrosion behavior; the obtention of chromium-enriched oxides considerably enhances the corrosion resistance [26]. 

Improving the corrosion resistance directly by modifying the SLM processing parameters has proven to be a difficult task, so researchers have been looking at other methods. Heat treatment is the most frequent approach found in the literature, in which the SLMed 316L SS components are submitted into a recrystallization post-process, in which dislocations in the grain boundaries disappear [17], and residual stress is relieved [27]. After the post-processing, the electrochemical properties are enhanced and, in some cases, the corrosion resistance of the SLMed 316L SS component might be better than a wrought component [19,22]. Reducing the porosity of SLMed components by heat treatment is of great importance as pores are the principal reason for the reduction in electrochemical and mechanical properties of an AM component. Trelewicz et al. [28] found a relationship between porosity and corrosion in AM alloys, in which crevice corrosion is mainly caused in the pores of the material. Likewise, Sun et al. [29] concluded that pre-existing pores in the SLMed 316L microstructure cause an increase in the metal dissolution, leading to pitting corrosion at low anodic potentials. On the other side of the spectrum, there are inclusions, cracks, and segregated elements that also affect the corrosion resistance of the SLMed parts. MnS inclusions influence the inhibition of pitting and its propagation over the material [30,31]. Mo segregation causes a loss of passivity and an increase in anodic current density [28].

In order to characterize the electrochemical characteristics of the SLMed 316L SS alloy, electrochemical and microscopy techniques need to be considered. Harun et al. [32] used the potentiodynamic polarization technique to study the corrosion behavior of the SLMed 316L SS exposed to thermal oxidation (TO) at controlled temperature (37 °C) in a Ringer’s solution. The thermal oxidation process (TO) was the key to find out different oxide layers (${\mathrm{Fe}}_{2}O_{3}$ and ${\mathrm{Cr}}_{2}O_{3}$) on the additively manufactured 316L SS alloy, where the oxide layer formed at a time of 150 h with 700 °C showed improved corrosion behavior. Chao et al. [30] performed a corrosion study using the SLMed 316L SS tested by the cyclic potentiodynamic polarization technique, whereby it was found that the passive range exhibited by the SLMed component is wider than that of wrought 316L SS. Herein, the large passive area as a good anticorrosive property in steel alloys [25] was correlated to a minimal amount of MnS inclusions found from microstructural characterization via scanning and transmission electron microscopy (SEM and TEM). In addition, Kong et al.’s study [17] previously mentioned considered nondestructive corrosion tests to evaluate the electrochemical response of SLMed 316L SS, which was subjected to heat treatment to increase its corrosion resistance. After the recrystallization process, electrochemical results verified an outstanding corrosion behavior, superior to wrought. 

It is also important to investigate the type of corrosion occurring in the SLM material. In fact, the most common corrosion studies of an SLM material focus on the obtention of the corrosion resistance rather than the relation of the surface and microstructure orientation with the type of corrosion occurring in the material for different corrosive environments. For this, electrochemical noise (EN) characterization appears to be a promising technology [33] to provide an understanding of the corrosion behavior produced in a metallic material, which would be difficult to understand by other means. Due to the material’s fluctuations in potential and current in the experiment, these can be represented by a corrosion map [34] thanks to the EN approach. The amplitude of such fluctuations can define the amount of the dissolved material during the corrosion process. Therefore, the aim of this work was to study the corrosion behavior of 316L SS samples manufactured via SLM using the EN technique and its microstructural features via optical microscopy (OM), scanning electron microscopy (SEM), and energy-dispersive X-ray spectroscopy (EDS), as well as X-ray fluorescence spectroscopy (XRF), to analyze the chemical composition of the material. The material was treated on two different surfaces, horizontal and vertical, in order to study how the microstructure direction influences the corrosion behavior.

## 2. Materials and Experimental Methods

### 2.1. Material Preparation

316L SS alloy powder with a spherical shape (10–45 μm), as shown in Figure 1, produced by LPW Technologies (Runcorn, UK) with the chemical composition (in wt.%) shown in Table 1, was chosen to manufacture cubic samples using the SLM process. The manufacturing of the samples for electrochemical corrosion tests and experimental observation took place in an SLM GmbH 280 HL machine (SLM Solutions, Lubeck, Germany) with the processing parameters shown in Table 2. The SLM process was performed in an argon-filled chamber. The SLM-prepared samples were also analyzed by X-ray fluorescence spectroscopy (XRF) (Olympus, Tokyo, Japan) to obtain their chemical composition, as shown in Table 1.

Note in Table 1 that the values of the chemical composition of the powder and the SLM-prepared sample do not agree very well. This could be because it could have generated certain agent contaminants inside the Ar atmosphere, which consequently are able to demote the chemical structure of the SLM material. It was observed that the SLM process studied in this research kept working with the oxygen levels up to 1% for certain periods of time, which means that a small loss of Ar gas existed during the laser process. Although the most logical is that the amount of Cr and Mo would have been in lower values, this does not occur in that way. Thus, important attention must be taken into account when the 316L SS alloy is processed by SLM and it is recommended that the oxygen levels inside the Ar atmosphere be completely less to 0.01% in order to not generate certain contamination issues in the material. 

As Figure 1 shows, measurements in some powder particles were performed in order to verify the particle size supplied by the manufacturer (LPW Technology), together with an EDS analysis on one of those particles, showing its elemental composition with the major alloying elements and having a similar variation with respect to the chemical composition of the powder presented in Table 1, i.e., the Cr and Mo are also incremented in values of 18.20 and 2.85 wt.%, respectively. As for the particle size, measurements in some powder particles were performed, as shown in Figure 1, taking an average of 10 particles in three different zones of the SEM image. 

Figure 2 shows the parts manufactured by SLM (Figure 2a), as well as the sample, in a digital form with a cutting plane to show how the samples were cut in order to obtain the cross-section of each specimen (Figure 2b). In this study, the XY and XZ (Z-axis is parallel to the build direction) planes of the samples were selected as the testing surfaces to carry out the electrochemical tests and verify how the surface orientation influences the final electromechanical properties. Although Figure 2b only shows the vertical cutting plane, some samples were also cut in a horizontal direction (XY plane), cutting all the specimens in half. 

The as-built samples were grinded and polished according to standard procedures, up to 4000 grits SiC abrasive paper. In accordance with these procedures, different working areas corresponding to the horizontal and vertical cross-section of the sample, normal (XY plane) and parallel (XZ plane) to the build direction, respectively, were obtained. As the SLMed 316L SS microstructure was also obtained in this work, some of the polished surfaces were etched using Vilella’s reagent (J.B. Baker, Monterrey, Mexico) (1 g of Picric acid, 5 mL of hydrochloric acid, (concentrated), and 95 mL of ethyl alcohol) for 6 min to be observed by a Zeiss axiobserver.Z1m optical microscope (Carl Zeiss, Oberkochen, Germany) to show the shape and microstructure of the melt pool (Section 3.2).

### 2.2. Microscopic Observation of the XY and XZ Testing Surfaces

In order to observe the initial features of the treated surfaces after the polishing process, a superficial observation using the optical microscope was performed on the horizontal and vertical surfaces. This optical observation was carried out for scanning the possible SLM defects that were developed during the laser process. For this, four magnifications in the optical microscope were taken into account to show a selected area of the testing surface, corresponding to the XY and XZ plane of the material. Figure 3 shows the metallographic images of the two analyzed surfaces, which were labeled with a and b, and c and d, for the horizontal and vertical surfaces, respectively. It can be observed that there existed porosity in form of irregularities, inclusions, irregular, and spherical defects. It is worth mentioning that a detailed analysis of the SLM irregularities is also considered in Section 3.1. For this, a ZEISS EVO scanning electron microscope (SEM) (Carl Zeiss, Oberkochen, Germany) was used, as well as to analyze and observe the corrosion products after the corresponding measurements, Section 3.4.

### 2.3. Corrosion Testing

The corrosion tests were carried out using a linear polarization resistance (LPR) [35] and electrochemical noise (EN) [36] experimental setup, as shown in Figure 4. A Gill-AC potentiostat/galvanostat/ZRA (Zero Resistance Ammeter) (ACM Instruments, Cumbria, United Kingdom) and a three-electrode cell integrated the test system, where a saturated calomel electrode (SCE), a platinum mesh, and the SLMed 316L SS sample made up the reference, counter, and work electrode, respectively.

Once the experimental setup was prepared, the SLMed 316L SS polished surface of each sample was exposed and studied in three electrolyte solutions, prepared with analytical grade reagents (J.B. Baker, Monterrey, Mexico); $H_{2}O$, 3.5 wt.% NaCl, and $20{\%\;H}_{2}{\mathrm{SO}}_{4}$. The linear polarization resistance (LPR) measurements were carried out with a sinusoidal AC voltage amplitude of ±10 mV at a scanning rate of 0.166 mV/s. The electrochemical noise (EN) measurements were sampled using 1024 points at a rate of 1 point per second. The exposed area of each sample was $1{\;\mathrm{cm}}^{2}$. All the electrochemical corrosion tests were performed at room temperature (20 ±2 °C). The final electrochemical properties of the samples immersed in each corrosive environment were an average of three measurements performed. In addition, it is also important to mention that the samples were stored in a desiccator (El Crisol, Monterrey, Mexico) after being polished with 4000 grits SiC abrasive paper for a time of 24 h before the electrochemical measurements. 

## 3. Results

### 3.1. Chemical Microanalysis

X-ray energy-dispersive spectroscopy (EDS) was also performed in order to characterize the morphology of the specimen’s defects before making the electrochemical measurements. For this, a selected area of the horizontal and vertical testing surface was explored in which imperfections with their respective shape and chemical composition were identified. Figure 5 shows two SEM micrographs displaying some irregularities with their corresponding chemical composition. In this case, Figure 5a presents an SLM defect of 10 μm longitude in the horizontal surface. Its elemental composition at a specific area inside of it, red zone in Figure 5b, showed high contents of Si and Mn, as well as O. The irregularity shown in Figure 5c with its respective chemical composition in Figure 5d presented high contents in Fe and O and was in the vertical surface of the material. The compositional analysis of the SLM defect shown in Figure 5a confirmed the results by Chao et al.’s [30] study, where it was reported that the SLMed 316L SS alloy is susceptible to content spherical and irregular defects with Si and O. In addition to this, it can be observed that the SLM defects shown in Figure 5a,c had a different shape with each other; i.e., in Figure 5a, the pore had an outstretched shape, and in Figure 5c, there were two types—spherical- and irregular-shaped pores. Although the SLM defect presented in Figure 5a was not completely spherical, it seems that it was formed from a spherical SLM defect. According to this, it can be established that the horizontal surface of an SLMed material suffered from shape-outstretched pores formed from spherical pores, while the vertical surface had spherical and irregular pores. It is well known that spherical pores are provoked by gas induced from the liquid metal during the fast solidification process, and irregular defects are associated with the lack of metallic powder [37].

Although EDS analyses were carried out at the microscopic level (SEM), these agree with the results of nano-inclusions (TEM) reported by Chao et al. [30], where the elemental composition of the nano-defects reported high contents of Si and Mn. In fact, Saeidi et al.’s [38] study pointed out that nano-inclusions are formed due to the accumulation of O, Al, Si, and Mn. Based on this, the SEM image of the horizontal surface shown in Figure 6 confirmed the same, i.e., certain impurities with high contents of Mn, Cr, and O were also localized, which could lead to the formation of (MnCr)O inclusions. Even though these defects cannot be considered as localized pitting sites, the Cr content accompanied by the high Si content could mean the formation of silicates, as reported by Kong et al. [17].

### 3.2. Characterization of the Microstructure

Figure 7 shows the microstructure of the SLMed 316L stainless steel. The micrograph was obtained by optical microscopy (OM), which shows the resulting micro/macro-structure of the horizontal and vertical surfaces, normal and parallel to the build direction, respectively. It can be observed from the shape of the melt pool for both surfaces and for certain points how grains tend to grow up from the fusion line to the center of the melt pool. It is well known that the growth direction to the melt pool center is due to the thermal gradients produced by the SLM process, which are higher in the melt pool center than along the fusion line [37,38,39]. In addition, the microstructure was characterized by martensitic with a dark brown tone, pointed out with the yellow arrow, whereas austenite is shown with a clearer brown tone and pointed out with the blue arrow. It can be observed from Figure 7a,b that the fusion lines or melt pool boundaries were generated by the Gaussian distribution of the laser energy density. Although the melt pool boundaries were not so distinguishable, it is possible to appreciate the shape of the melt pool for each metallographic image. Figure 7a shows the melt pools with the shape of enlarged ovals located at 90° due to the scan rotation between the successive layers (SLM processing parameter mentioned in Table 2), and Figure 7b illustrates the cross-section of the melt pool with a semicircular shape, typical of the SLM process. 

It is important to notice how the powder melting manufacturing process can produce fine grains that could influence the final electromechanical characteristics of the material. As for the SLM microstructure in the horizontal surface (XY plane) of the sample, it has been established that fusion lines could not be completely true. According to Rao et al. [23], this is because the melt pool cores are not totally normal to the observation XY plane, with the samples being subjected to the cutting and polishing process. Nevertheless, the SLM microstructures of this research work were taken carefully, based on the standard procedures. It is important to highlight that the size of the melt pool (melt pool width and depth) was not considered in the influence on the corrosion behavior.

### 3.3. Corrosion Results

#### 3.3.1. Linear Polarization Resistance (LPR)

Corrosion rate measurements of the SLMed samples were calculated using the Stern–Geary relationship [40], Equation (1):(1)$$I_{corr}=\frac{\beta }{\Delta E/\Delta I}=\frac{\beta }{R_{p}}$$
where β is the Stern–Geary constant, ΔE is the potential gradient, ΔI is the current gradient, and ΔE/ΔI is the polarization resistance ($R_{p}$). Note that in Equation (2), the Stern–Geary constant can be calculated from the anodic ($b_{a}$) and cathodic $b_{c}$ Tafel slopes [41].
(2)$$\beta =\frac{b_{a}b_{c}}{2.3\left(b_{a}+b_{c}\right)}$$

Based on the LPR technique, the SLMed 316L SS was studied in the three electrolytes, and its corrosion characteristics ($E_{corr}$, $I_{corr}$, and corrosion rate) were obtained. First, the horizontal samples were considered for the electrochemical measurements. For this, Table 3 shows the corresponding parameters, where it can be observed that the major corrosion rate is $1.31\times 10^{-3}\;\mathrm{mm}/\mathrm{year}$ obtained in $H_{2}{\mathrm{SO}}_{4}$ in comparison with the value of $8.85\times 10^{-5}\;\mathrm{mm}/\mathrm{year}$ calculated in NaCl, as well as with the value of $7.8\times 10^{-5}\;\mathrm{mm}/\mathrm{year}$ for the $H_{2}O$ electrolyte.

On the other hand, the electrochemical parameters were also obtained for the vertical samples. For this, Table 4 shows the corrosion rate values of $9.2\times 10^{-5}$, $2.45\times 10^{-3}$, and $2.4\times 10^{-2}\;\mathrm{mm}/\mathrm{year}$, which correspond to the $H_{2}O$, NaCl, and $H_{2}{\mathrm{SO}}_{4}$ electrolytes, respectively. Note that the major corrosion rate is $2.4\times 10^{-2}\;\mathrm{mm}/\mathrm{year}$ obtained in $H_{2}{\mathrm{SO}}_{4}$ and the minor is $9.2\times 10^{-5}\;\mathrm{mm}/\mathrm{year}$ calculated in $H_{2}O$. As it can be noticed, the results between the two samples indicate that the corrosion rates for the XZ samples were higher than those of the XY samples. These results give a first notion of how the surface orientation of an SLM material influences its corrosion resistance.

#### 3.3.2. Electrochemical Noise (EN)

Electrochemical noise (EN) tests were also performed in this research. Although linear polarization resistance (LPR) gave a prediction about the corrosion resistance of the SLMed 316L SS, it is difficult to understand the type of corrosion. For this, Figure 8a shows the potential time series of the sample in the experiment when it was immersed in a chloride-containing solution at room temperature (20 °C ± 2). Figure 8b shows the current time series of the samples. Herein, it can be noticed that the XZ sample seemed to be more active than the XY sample. This is because the corrosion current density of the vertical sample increased rapidly for certain periods of time, even recording a few transients of high frequency and amplitude compared with the horizontal sample, as shown by the red line. According to Gaona et al. [34], the characteristics of those transients can lead to localized corrosion. On the other hand, it can be noticed that the horizontal sample maintained a constant potential during the test. Although there were also some transients in the XY sample, the stability in its corrosion potential indicates that it could have been generated by a passive film formed by chromium oxide, even after the cutting process of the samples. Additionally, the surface orientation also influenced the corrosion behavior and the film oxide composition [23,42]. In fact, the corrosion film formed naturally after cutting generated a small barrier to the aggressive agents of the electrolytes such as Cl^−^ ions. Nevertheless, once the system began to interact, this passive film had a path to the dissolution of metal ions (M^+^) in the anodic zones, while in the cathodic zones, the reduction of oxygen was carried out through the formation of OH^−^ ions.

Andreatta et al. [43] reported that the 316L SS alloy evaluated in NaCl with the acid and neutral environment presents a passivate behavior wider than the alloys fabricated conventionally. Besides, the SLMed 316L SS alloys do not present a tendency to metastable pitting [43]. Sander et al. [44] determined that the typical frequency of metastable pitting in 3D-printed 316L samples is, in many cases, considerably lower than that of the forged 316L. In addition, those with the highest frequency of metastable pitting are associated with higher porosity. The transients present in these samples are typical of stable pitting behavior.

Figure 9 shows the corrosion behavior of the SLM material developed by the electrochemical noise (EN) test using $H_{2}{\mathrm{SO}}_{4}$. In this case, the horizontal sample also tried to maintain a constant potential while the vertical sample increased its potential by several periods of time and then decreased throughout time. In Figure 9b, it seems there were a few transients in the current that could mean pitting corrosion for both samples. 

Figure 10a,b shows the time series in potential and current, respectively, of the horizontal and vertical samples immersed in $H_{2}O$. Note that the electrochemical response of the XZ sample marks several transients that could mean metastable pitting typical of these alloys [44] (Figure 10b); fortunately, its corrosion potential remained constant (Figure 10a). In the case of the XY sample, this slightly increased its corrosion potential and tended to passivate due to its constant corrosion current density. However, the vertical sample seemed to be more active than the horizontal sample as it had higher transients.

In order to summarize, Table 5 shows the electrochemical noise (EN) characteristics such as electrochemical noise resistance ($R_{n}$), corrosion current density ($I_{corr}$), corrosion rate, skewness, and the corrosion type of the samples in the experiment when they were immersed in the three electrolytes. It can be noticed that all the samples (horizontal and vertical) led to two types of corrosion, localized for NaCl, and uniform for the $H_{2}{\mathrm{SO}}_{4}$ and $H_{2}O$ electrolytes. It can be noted that the minor corrosion rate was $1\times 10^{-5}\;\mathrm{mm}/\mathrm{year}$ for the $H_{2}O$ electrolyte. The identification of the corrosion type is based on statistical moments with skewness, which are the 3rd and 4th statistical moments [45].

In Figure 11, a histogram is illustrated in order to compare the corrosion rate between the two samples (horizontal and vertical), taking into account the three electrolytes. In effect, it can be observed that the minor corrosion rate was presented in the XY sample, immersed in $H_{2}O$. On the contrary, the sample with the worst corrosion behavior was also horizontal but immersed in $H_{2}{\mathrm{SO}}_{4}$ as it had a major corrosion rate ($3.2\times 10^{-3}$ mm/year) according to the graph. It is worth mentioning that the obtained results of the corrosion current densities were an average of the anodic and cathodic reactions registered on the surface of the material immersed in the corresponding electrolyte throughout the testing time.

### 3.4. Corroded Surfaces of the SLMed Samples

Figure 12 shows the optical microscope images after the electrochemical tests, in which a localized attack was observed in the NaCl solution, as a product of the Cl ions present. In the electrolyte of $H_{2}O$ and $H_{2}{\mathrm{SO}}_{4}$ the corrosion mechanism was uniform; however, the acid generated a greater dissolution of the oxide layer formed by the AM 316L stainless steel.

In Figure 13, the morphologies taken via SEM and EDAX analysis of the XY and XZ specimens evaluated in NaCl are shown. It is observed that the pits were developed within the subgrains due to the galvanic par that developed between the Mo-enriched phases (subgrain boundaries) that were segregated during the fabrication due to the rapid solidification, which does not give time to the Mo atoms to complete the diffusion [46,47] and the Fe-enriched phases.

Through EDAX analysis, the presence of the element chlorine can be observed for both specimens in the XY and XZ planes, which will preferentially attack the subgrains that act as anodes and the grain boundaries enriched in Mo as the cathode [28]. In addition, the presence of the elements of the base material (chromium, nickel, molybdenum, and manganese) and the traces of elements that correspond to the test electrolyte are also influenced.

The corrosion process of the 316L AM alloys in the $H_{2}{\mathrm{SO}}_{4}$ electrolyte presented a very different behavior from that observed in a saline environment. In this case, for the specimens in the XY plane, the corrosion process was generated over the entire surface of the sample, and in some cases, it developed within the imperfections of the material, developed by the manufacturing process, specifically in the pore walls. The EDAX analyses showed the presence of sulfur, which, together with the acidic pH of the electrolyte, developed the corrosion process; the corrosion products were not identified in the imperfection core, which indicates that the process developed through the walls (Figure 14).

## 4. Discussion

It was found that the SLMed 316 L SS alloy presented a small increment in the corrosion rate with a vertical orientation compared with the horizontal direction. This was proved by LPR when the material was immersed in the three electrolytes. It can be supposed that the variation in the corrosion resistance was because of the surface orientation of the SLM parts. Besides, based on the EDS analyses performed in the SLM defects, these indicate that there were high concentrations of O, Al, Si, and Mn in the analyzed irregular and spherical pores, which could cause a loss of passivity and an increment in the corrosion current density, as it was observed in the vertical samples. 

Furthermore, as for the material tested by the EN technique, this method shows that the SLMed 316L SS alloy also had a corrosion resistance higher in the horizontal direction than for the vertical one when the material was tested in $H_{2}O$. On the contrary, it was lower in NaCl and $H_{2}{\mathrm{SO}}_{4}$ for the XY plane due to the spherical pores, which developed localized corrosion for the NaCl electrolyte. It can be observed that the localized attack, as a result of Cl ions, was presented in the pore walls rather than in the cores. Nevertheless, it can be established that the surface orientation of an SLM material did play an important role in its corrosion resistance when there were no severe SLM defects. In fact, pits were developed within the subgrains due to the galvanic par that developed between the Mo-enriched phases (subgrain boundaries).

It is worth mentioning that the EN technique, as well as the morphology observation via EDS, SEM, and XRF analyses, gave important considerations of AM parts processed by SLM. In fact, the experimental morphology observation indicated that the horizontal surface suffered from spherical gas-induced pores, while the vertical surface presented spherical and irregular pores associated with the lack of metallic powder during the laser process. As for the EN technique, this one made a first calculation of the type of corrosion developed for an additively manufactured material considering three different environments as the corrosion studies in the literature have only focused on corrosion resistance.

However, it still does not have a quantifiable factor, e.g., the grain size in the vertical and horizontal directions, levels of porosity in both surfaces, and size of the melt pool, that can quantitatively compare the best corrosion behavior in the horizontal samples. This study opens the door to study the SLMed 316L SS corrosion behavior from a microstructural viewpoint for future investigations. Besides, the corrosion resistance of SLMed 316L SS products can be studied considering the amount of chromium on the surface of the material.

## 5. Conclusions

Electrochemical properties of the SLMed 316L SS were studied using EN measurements and SEM analyses. In the research, the following remarks were found:
The SEM and EDS analyses performed on the polished surface of the samples indicated the formation of irregular and spherical pores with contents of O, Al, Si, and Mn in the SLM defect. The spherical pores were attributed to the induced gas during the fast solidification process, and irregular defects with dimensions of approximately 18 um were associated with the lack of metallic powder. It was confirmed by the EDS analyses of this research work that inclusions in the SLMed 316L SS were formed due to the accumulation of O, Al, Si, N, and Mn.Using the electrochemical technique of RPL, it was found out that the best corrosion behavior was developed in the horizontal samples than for the vertical ones. This could be because according to the tests, the SLMed 316L surfaces developed an initial passive-active behavior due to a corrosion film formed after the cutting process. At the same time, this oxide layer could form due to the interaction of the material with chloride-containing and acid solutions, similar to the oxide layers (${\mathrm{Fe}}_{2}O_{3}$ and ${\mathrm{Cr}}_{2}O_{3}$) formed through the thermal oxidation (TO) process on an SLMed 316L SS in the literature. Through the electrochemical noise (EN) technique, the −1.65 and −0.012 skewness factors were calculated, which indicates that the SLMed specimens developed localized corrosion in NaCl, for the horizontal and vertical samples, respectively. The localized corrosion associated with the Cl ions attacked mainly the subgrains of the microstructure and, in some cases, defects such as pores. Although the initiation of metastable pits was found due to some pre-existing SLM defects (pores) on surfaces, it could not transition to stable growing pits, thanks to the corrosion film formed in the material. For the $H_{2}O$ and $H_{2}{\mathrm{SO}}_{4}$ electrolytes, the corrosion type was uniform for the two orientations, due to it developing over the entire surface of the samples. Besides, according to the EN parameters, it was determined that the best corrosion rate in $H_{2}O$ was $1\times 10^{-5}\;\mathrm{mm}/\mathrm{year}$ for the horizontal samples compared with the value of $7\times 10^{-4}\;\mathrm{mm}/\mathrm{year}$ for the vertical ones. However, the XY samples presented a minor corrosion resistance than for the XZ samples both tested in the NaCl and $H_{2}{\mathrm{SO}}_{4}$ electrolytes.The morphology identified by SEM was correlated with the results obtained from the electrochemical techniques, where it was observed that the pits developed within the subgrains due to the galvanic par that developed between the Mo-enriched phases (subgrain boundaries) that were segregated during the fabrication.

## Figures and Tables

**Figure 1 materials-14-04509-f001:**
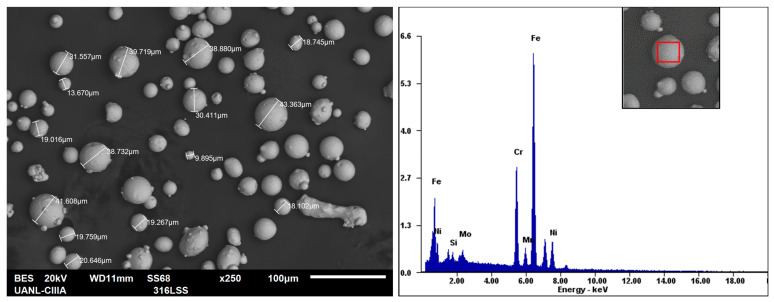
SEM image of the 316L stainless-steel metal powder obtained in this study showing the size of some particles (**left side**) and EDS analysis in one of those particles (**right side**).

**Figure 2 materials-14-04509-f002:**
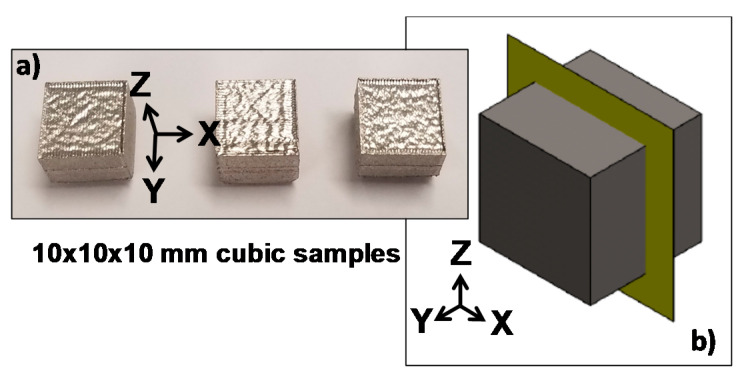
(**a**) Cubic samples prepared by the SLM process. (**b**) Cutting plane used as a reference to obtain the cross-section of each sample.

**Figure 3 materials-14-04509-f003:**
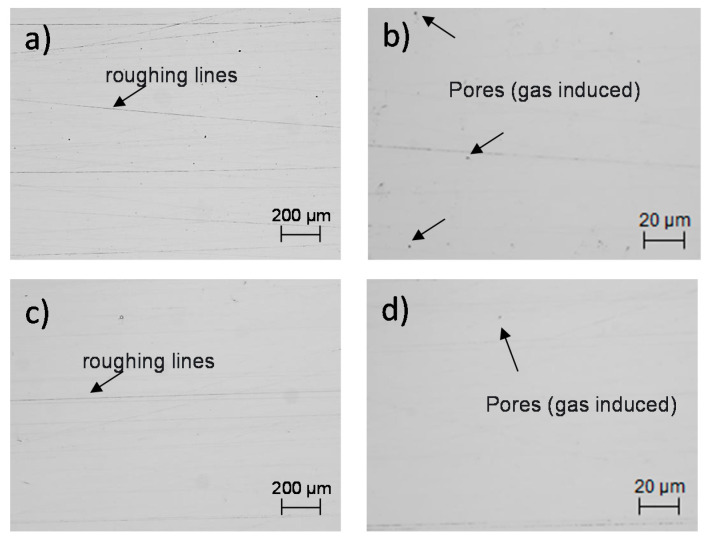
Metallographic images of the SLMed 316L SS horizontal and vertical surfaces showing porosity. Images: (**a**) (50X) and (**b**) (500X) XY plane, and (**c**) (50X) and (**d**) (500X) XZ plane.

**Figure 4 materials-14-04509-f004:**
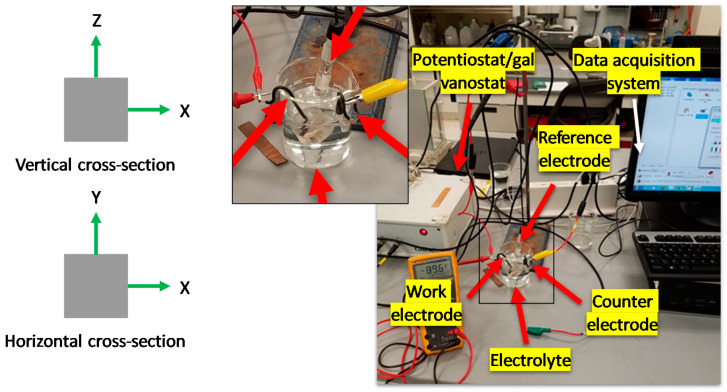
Experimental setup used to perform the corrosion tests; left side, and horizontal and vertical cross-sections (Z axis is parallel to the build direction) of the work electrodes.

**Figure 5 materials-14-04509-f005:**
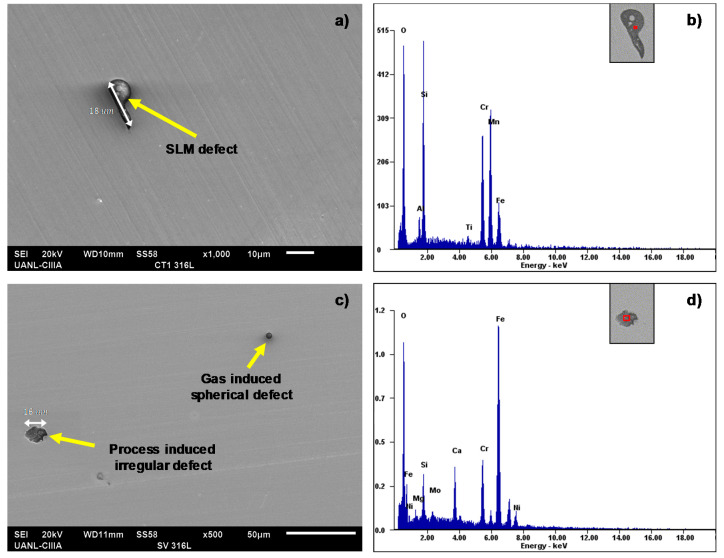
SEM images of the polished surfaces of the samples displaying some SLM defects in the material (left-hand side) (**a**,**c**; 100X and 500X, respectively) and chemical composition obtained by EDS of those irregularities (right-hand side; **b**,**d**).

**Figure 6 materials-14-04509-f006:**
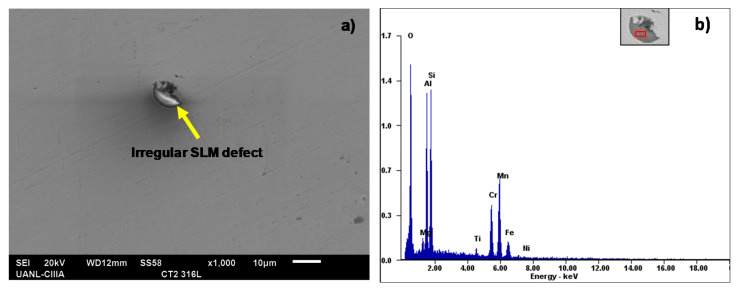
(**a**) SLM inclusion found on the polished surface and (**b**) its EDS chemical composition showing high contents of O, Al, Si, and Mn.

**Figure 7 materials-14-04509-f007:**
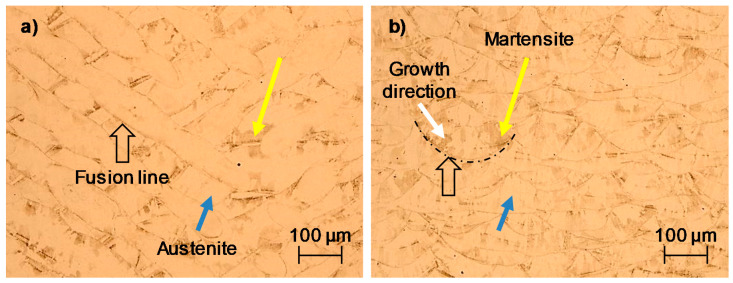
Representative optical micrograph showing the SLMed 316L SS microstructure of the horizontal (**a**) and vertical (**b**) surfaces. Growth direction is indicated with a white arrow while the fusion line is shown with an open arrow.

**Figure 8 materials-14-04509-f008:**
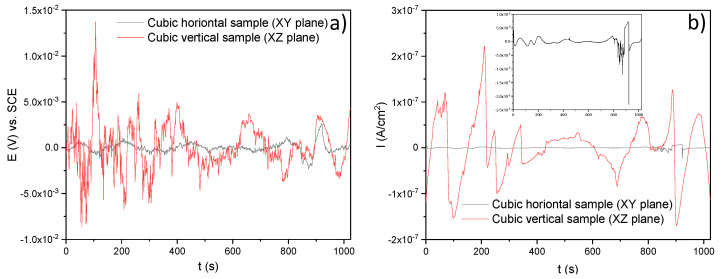
Time series in potential and current of the SLMed 316L SS in 3.5 wt.% NaCl: (**a**) potential versus time and (**b**) current versus time.

**Figure 9 materials-14-04509-f009:**
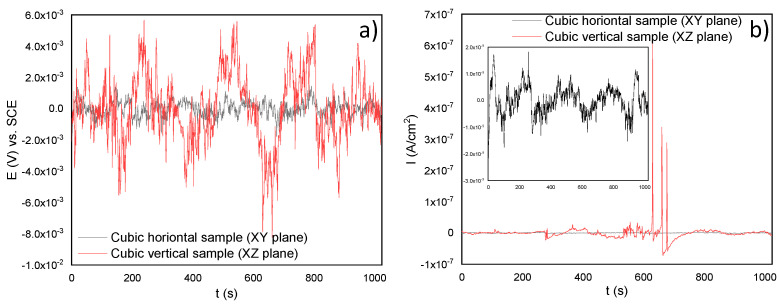
Time series in potential and current for the SLMed 316L SS in $20{\%\;H}_{2}{\mathrm{SO}}_{4}$: (**a**) potential versus time and (**b**) current versus time.

**Figure 10 materials-14-04509-f010:**
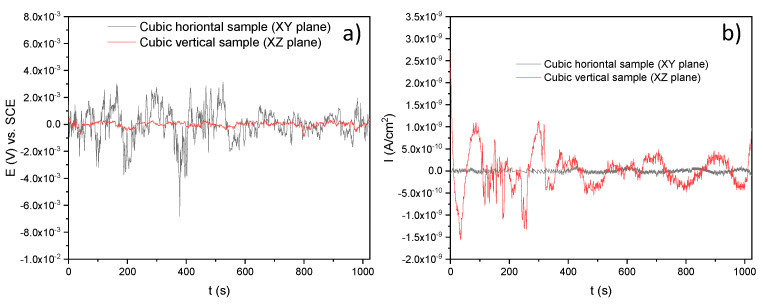
Times series in potential and current for the SLMed 316L SS in $H_{2}O$: (**a**) potential versus time and (**b**) current versus time.

**Figure 11 materials-14-04509-f011:**
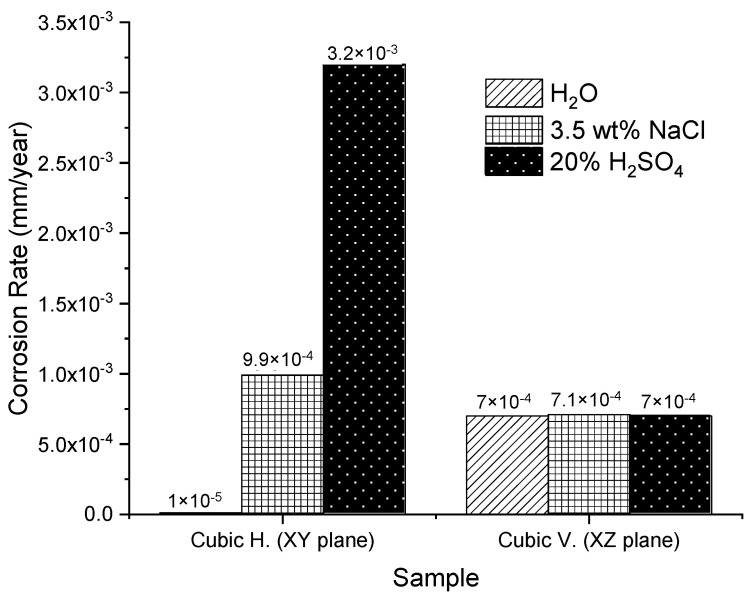
Comparison of the corrosion rate between the samples in the three electrolytes. The number of points for each corrosion test was 1024 points at a rate of 1 point per second. The average corrosion rate for each performed test is shown.

**Figure 12 materials-14-04509-f012:**
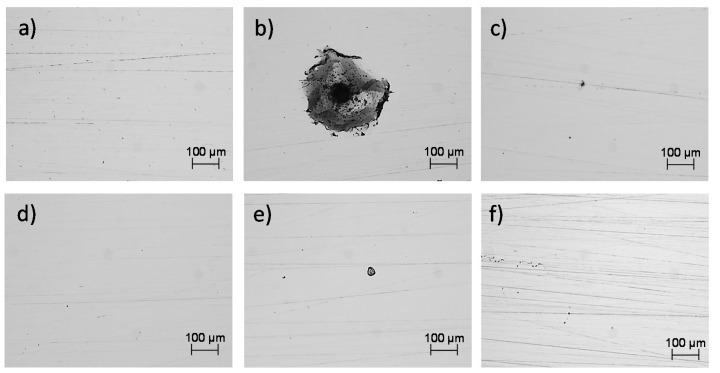
Morphologies of the specimens after the electrochemical tests. Optical microscope at 100X. XY plane: (**a**) $H_{2}O$, (**b**) NaCl, and (**c**) $H_{2}{\mathrm{SO}}_{4}$; XZ plane: (**d**) $H_{2}O$, (**e**) NaCl, and (**f**) $H_{2}{\mathrm{SO}}_{4}$.

**Figure 13 materials-14-04509-f013:**
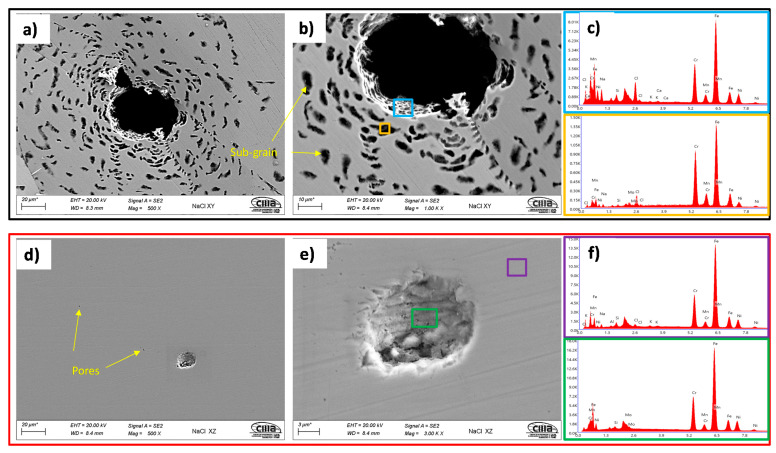
Morphologies of the specimens after the electrochemical tests in 3.5 wt.% NaCl at 500X and 1000X. XY plane (**a**–**c**); XZ plane (**d**–**f**). ZEISS EVO scanning electron microscope (SEM).

**Figure 14 materials-14-04509-f014:**
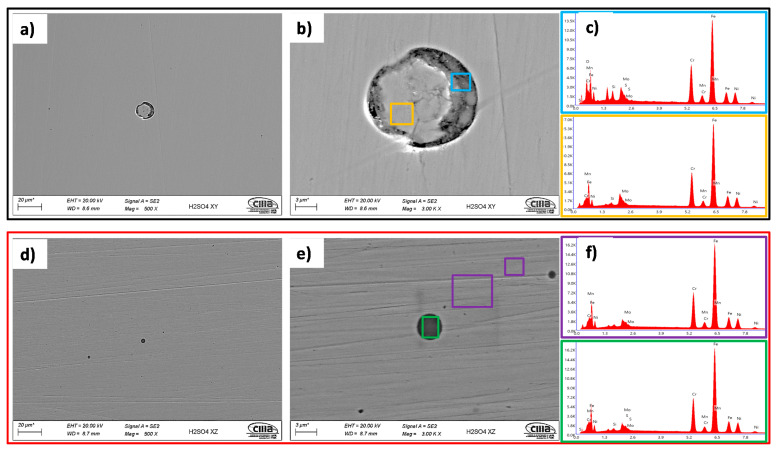
Morphologies of the specimens after the electrochemical tests in $H_{2}{\mathrm{SO}}_{4}$ at 500X and 1000X. XY plane (**a**–**c**); XZ plane (**d**–**f**). ZEISS EVO scanning electron microscope (SEM).

**Table 1 materials-14-04509-t001:** Chemical composition (in wt.%) of the 316L SS powder and the SLM-prepared sample.

Element	C	Cr	Cu	Mn	Mo	Ni	P	S	Si	Fe
Powder	0.03	17.5–18.0	0.50	2.0	2.25–2.50	12.5–13	0.025	0.01	0.75	Bal.
SLM-prepared sample	-	18.67	0.04	1.13	2.55	11.84	-	-	-	Bal.

**Table 2 materials-14-04509-t002:** SLM processing parameters used to manufacture the 316L SS samples.

Parameter	Value
Laser power (W)	400
Scanning speed (mm/s)	230
Hatch spacing (μm)	110
Layer thickness (μm)	30
Scan rotation between successive layers	90°
Energy density (J/mm^3^)	527

**Table 3 materials-14-04509-t003:** Electrochemical parameters of the SLMed 316L SS tested in horizontal (XY plane).

Sample	Electrolyte	Electrochemical Parameters		
		$E_{corr}\;vs.\;SCE$ (mV)	$I_{corr}$ $\left(mA/cm^{2}\right)$	Corrosion Rate (mm/year)
Horizontal (XY plane)	$H_{2}O$	−123± 6	$7.29\times 10^{-6}$ $\pm \;4\times 10^{-7}$	$7.8\times 10^{-5}$ $\pm \;4\times 10^{-6}$
	3.5 wt.% NaCl	22± 1	$8.27\times 10^{-6}$ $\pm \;4\times 10^{-7}$	$8.85\times 10^{-5}$ $\pm \;4\times 10^{-6}$
	$20{\%\;H}_{2}{\mathrm{SO}}_{4}$	112± 5	$1.23\times 10^{-4}$ $\pm \;6\times 10^{-6}$	$1.31\times 10^{-3}$ $\pm \;6\times 10^{-5}$

**Table 4 materials-14-04509-t004:** Electrochemical parameters of the SLMed 316L SS tested in vertical (XZ plane).

Sample	Electrolyte	Electrochemical Parameters		
		$E_{corr}\;vs.\;SCE$ (mV)	$I_{corr}$ $\left(mA/cm^{2}\right)$	Corrosion Rate (mm/year)
Vertical (XZ plane)	$H_{2}O$	23± 1	$8.6\times 10^{-6}$ $\pm \;4\times 10^{-7}$	$9.2\times 10^{-5}$ $\pm \;5\times 10^{-6}$
	3.5 wt.% NaCl	−108± 5	$2.3\times 10^{-4}$ $\pm \;1\times 10^{-5}$	$2.45\times 10^{-3}$ $\pm \;2\times 10^{-3}$
	$20{\%\;H}_{2}{\mathrm{SO}}_{4}$	278 ± 14	$2.3\times 10^{-3}$ $\pm \;1\times 10^{-4}$	$2.4\times 10^{-2}$ $\pm \;1\times 10^{-3}$

**Table 5 materials-14-04509-t005:** Electrochemical noise (EN) characteristics of the SLMed 316L SS tested in the three electrolytes (3.5 wt.% NaCl, $20{\%\;H}_{2}{\mathrm{SO}}_{4}$, and $H_{2}O)$.

Electrolyte		Sample	
		Horizontal	Vertical
		(XY Plane)	(XZ Plane)
3.5 wt.% NaCl	$R_{n}\;\left(\Omega -{\mathrm{cm}}^{2}\right)$	299,514	41,451
	$I_{corr}\;\left(\mathrm{mA}/{\mathrm{cm}}^{2}\right)$	$8.7\times 10^{-5}\;\pm \;4\times 10^{-6}$	$6.27\times 10^{-5}\;\pm \;4\times 10^{-6}$
	Corrosion rate (mm/year)	$9.9\times 10^{-4}\;\pm \;5\times 10^{-5}$	$7.1\times 10^{-4}\;\pm \;4\times 10^{-5}$
	Skewness	−1.65	−0.012
	Corrosion type	Localized	Localized
$20{\%\;H}_{2}{\mathrm{SO}}_{4}$	$R_{n}\;\left(\Omega -{\mathrm{cm}}^{2}\right)$	926,782	422,588
	$I_{corr}\;\left(\mathrm{mA}/{\mathrm{cm}}^{2}\right)$	$2.8\times 10^{-4}\;\pm \;1\times 10^{-5}$	$6.15\times 10^{-5}\;\pm \;6\times 10^{-6}$
	Corrosion rate (mm/year)	$3.2\times 10^{-3}\;\pm \;2\times 10^{-4}$	$7\times 10^{-4}\;\pm \;4\times 10^{-5}$
	Skewness	−0.17	0.27
	Corrosion type	Uniform	Uniform
$H_{2}O$	$R_{n}\;\left(\Omega -{\mathrm{cm}}^{2}\right)$	29,136,364	422,588
	$I_{corr}\;\left(\mathrm{mA}/{\mathrm{cm}}^{2}\right)$	$8.9\times 10^{-7}\;\pm \;4\times 10^{-8}$	$6.15\times 10^{-5}\;\pm \;3\times 10^{-6}$
	Corrosion rate (mm/year)	$1\times 10^{-5}\;\pm \;5\times 10^{-7}$	$7\times 10^{-4}\;\pm \;4\times 10^{-5}$
	Skewness	0.05	0.27
	Corrosion type	Uniform	Uniform

## Data Availability

The data presented in this study are available on request from the corresponding author. The data presented is contained within the article.

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
