# Peer review of "Corrosion Resistance Measurement of 316L Stainless Steel Manufactured by Selective Laser Melting"

_materials, 2021, doi:10.3390/ma14164509_

Round 1
Reviewer 1 Report
Corrosion resistance of 316ss manufactured by selective laser melting method was investigated through the linear polarization resistance measurement. In particular, the surface dependency on corrosion behavior was discussed, which might be relating to manufacturing process influenced by laser irradiation direction. The following items should be clear or described in more detailed to understand contents mentioned in the paper.
- As mentioned in the conclusion, the corrosion behavior is very dependent on the cutting surface from the cubic samples. The more detailed manufacturing process should be described from the viewpoint of the micro-structure formation, which is dependent on the vertical and horizontal directions, and the laser irradiation direction.
- Illustrate the size of the cubic samples in Fig.2.
- Describe how much the difference dependent on the vertical and horizontal is there ; i.e. grain boundary length, coverage of Austenite and Martensite, etc., and describe numerically.
- Show the microstructures of the vertical and horizontal surfaces.
- Discuss on how the difference of the microstructures occurs; i.e. laser irradiation direction, gravity direction, thermal diffusion, etc.
- Show the corroded surface after corrosion tests. Discuss on the relationship between the microstructure before and after cession tests.
- As for the Fig.10, indicate the total corrosion time and mention the average corrosion rate throughout the corrosion tests.
- Already passive film was formed just after cutting the vertical and horizontal surfaces. It maybe influences on the corrosion attack in the case of NaCl with the small corrosion rate.
- Show the temperature of each corrosive liquid during the corrosion tests, which is very important information for corrosion behavior.
Reviewer 2 Report
Abstract
The purpose of the study as well as findings should be better, e.g. more explicitly, articulated.
Introduction
The aim of the study should be clearly indicated.
The motivation, novelty and importance of your study should be highlighted.
Since you discuss effects between surface morphology and corrosion, it requires to do a short review on how surface texture influence these kind of processes. See for instance:
https://doi.org/10.1016/j.apsusc.2020.146838
and
10.1016/j.materresbull.2017.03.022
The potential phenomena that stand behind different corrosion resistance of additively manufactured parts with respect to their conventionally manufactured counterparts should be carefully discussed in this review.
Materials and methods:
Why do chemical composition of powder is different that SLMed samples – see table 1.
How the samples were cut? How does the cutting influence the corrosion performance?
Description of corrosion process should come last in this section as you present some results before the corrosion in the same section which is confusing.
In the abstract, it is mentioned that you optical microscopy but nothing is presented in this section.
Some results are presented in this section while they need to presented in Results section
Results
Section 3.1. is too brief and has to be expanded
Some new analysis are introduced while this section should only present the results not the new methods.
The results shows some differences in corrosion between samples as cut with differently oriented planes. However, they do not mention any quantifiable factor which influence the varied performance. Nothing about the surface nor microstructure is shown using numbers! I would strongly expected a full quantitative analysis.
Discussion:
This part is non-existent! The authors should discuss the importance, novelty and limitation of their study. They should explain why there is a difference in their results and refer to already published works. Some information about future work is also needed.
Round 2
Reviewer 2 Report
Thank you very much for improving your paper.
